# Stunting as a Synonym of Social Disadvantage and Poor Parental Education

**DOI:** 10.3390/ijerph18031350

**Published:** 2021-02-02

**Authors:** Christiane Scheffler, Michael Hermanussen, Sugi Deny Pranoto Soegianto, Alexandro Valent Homalessy, Samuel Yan Touw, Sevany Isabella Angi, Queen Sugih Ariyani, Tjahyo Suryanto, Giovanni Kathlix Immanuel Matulessy, Taolin Fransiskus, Andrea V. Ch. Safira, Maria Natalia Puteri, Rani Rahmani, Debora Natalia Ndaparoka, Maria Kurniati Ester Payong, Yohannes Dian Indrajati, Reynardo Kurnia Hadiyanto Purba, Regina Maya Manubulu, Madarina Julia, Aman B. Pulungan

**Affiliations:** 1Institute of Biochemistry and Biology, Human Biology, University of Potsdam, 14469 Potsdam, Germany; 2University of Kiel, Aschauhof, 24340 Eckernförde-Altenhof, Germany; michael.hermanussen@gmail.com; 3Indonesian Pediatric Society, East Nusa Tenggara Branch, Kupang-East Nusa Tenggara, Kupang 85351, Indonesia; s.d.p.soegianto@gmail.com (S.D.P.S.); sunotnay_qt@yahoo.co.id (T.S.); franstaolin01@yahoo.com (T.F.); grandiangi21@gmail.com (R.M.M.); 4Indonesian Medical Association Branch, Kupang 85351, Indonesia; alexandrohomalessy@gmail.com (A.V.H.); samytouw@gmail.com (S.Y.T.); queen.ariyani@gmail.com (Q.S.A.); giomatulessy0196@gmail.com (G.K.I.M.); fira.christa@gmail.com (A.V.C.S.); marianataliaputri97@gmail.com (M.N.P.); ranirahmani04@gmail.com (R.R.); debora.slalutersenyum@yahoo.com (D.N.N.); esterpayong@gmail.com (M.K.E.P.); 5DDS PPDGS Konservasi Gigi FKG UGM, Jl. Sekip Utara, Sumatra, Medan 20153, Indonesia; stevaniisabell@gmail.com (S.I.A.); yohannesdian@gmail.com (Y.D.I.); 6Faculty of Medicine, Nusa Cendana University, Kupang 85351, Indonesia; purbareynardo@gmail.com; 7Department of Child Health, Faculty of Medicine, Public Health and Nursing, Universitas Gadjah Mada-Dr. Sardjito General Hospital, Yogyakarta 55281, Indonesia; madarinajulia@yahoo.com; 8Department of Child Health, Faculty of Medicine, Universitas Indonesia-Ciptomangunkusumo Hospital, Jakarta 10430, Indonesia; amanpulungan@mac.com

**Keywords:** stunting, social, economic, political and emotional factors on growth

## Abstract

Socially, economically, politically and emotionally (SEPE) disadvantaged children are shorter than children from affluent background. In view of previous work on the lack of association between nutrition and child growth, we performed a study in urban schoolchildren. We measured 723 children (5.83 to 13.83 years); Kupang, Indonesia; three schools with different social background. We investigated anthropometric data, clinical signs of malnutrition, physical fitness, parental education, and household equipment. Subjective self-confidence was assessed by the MacArthur test. The prevalence of stunting was between 8.5% and 46.8%. Clinical signs of under- or malnutrition were absent even in the most underprivileged children. There was no delay in tooth eruption. Underprivileged children are physically fitter than the wealthy. The correlation between height and state of nutrition (BMI_SDS, skinfold_SDS, MUAC_SDS) ranged between r = 0.69 (*p* < 0.01) and r = 0.43 (*p* < 0.01) in private school children, and between r = 0.07 (ns) and r = 0.32 (*p* < 0.01) in the underprivileged children. Maternal education interacted with height in affluent (r = 0.20, *p* < 0.01) and in underprivileged children (r = 0.20, *p* < 0.01). The shortness of SEPE disadvantaged children was not associated with anthropometric and clinical signs of malnutrition, nor with delay in physical development. Stunting is a complex phenomenon and may be considered a synonym of social disadvantage and poor parental education.

## 1. Introduction

Stunting is not a synonym of malnutrition. In a previous Indonesian study [1] we had started to question the concept of stunting as *prima facie* evidence of malnutrition and chronic infection. The rational of the present study was to further scrutinize current explanatory concepts of stunting and delineate social and emotional factors that are associated with poor child growth under low income conditions.

There is no argument that caloric shortage delays physical growth. Abundant literature describes the effect of acute and chronic malnutrition on children during intermittent periods of war and post-war economic disaster. Child growth is a responsive and very dynamic process and closely mirrors environment and economic conditions. But to the same extent, growth is impaired during periods of food shortage, it quickly recovers with marked catch-up whenever the growth limiting conditions terminate. Catch-up growth is characterized by height velocity above the limits of normal for age after a period of growth impairment [2] Catch-up growth is cause-specific as it occurs when the reason for the preceding growth impairment has been removed [3]. Already in 1919 Schlesinger [4] stated that “The whole disturbance of growth described here (children suffering from food shortage during World War One) is to be regarded as a simple inhibition; the growth type, the growth curve did not undergo any significant change in its form, apart from the occasionally observed slight delay of the puberty drive, the onset of the pubertal rise. The fact of inhibition of the longitudinal growth of the children during the last years of the war is of more scientific interest than practical importance for the above mentioned reason of the expected reparative capacity; it is an indication of the intensity of the general malnutrition of the children.” Witnessing the catch-up in growth when conditions had improved, he concluded that “the child’s longitudinal growth is largely independent of the extent and nature of the diet.” Similar statements were frequently published in these days [5].

Reasons for poor child growth are manifold. Already a century ago, pediatricians highlighted the association between economic affluence, social strata and growth, and explicitly questioned any prominent role of nutrition. The German paediatrician Meinhard von Pfaundler [6] stated 1916 “this question (short stature of the socially disadvantaged children) is important when considering the reason for the undersize of the poor. Malnutrition has often been mentioned... but, it does not seem appropriate to me to address malnutrition as the decisive factor in the undersize of the poor.” Pfaundler [6] particularly emphasized, that the body weight of those is not reduced to that extent that the reduction in body length would suggest, but less. The differences are not great, but quite constant. Other authors of the early 20th century published similar observations [5], and later, also pointed to the importance of emotional deprivation and its effect on infant and child growth [7].

The association between stunting and poor parental education is also well documented, even though it appears to have weakened recently [8]. Mensch et al. [9] reviewed evidence for a causal link between education and health, similar to what we had observed in our previous study [10].

In view of this evidence, we performed a second anthropometric study in urban schoolchildren from Kupang, West-Timor, Indonesia. Kupang is the capital of the Indonesian province of East Nusa Tenggara, with an estimated population in 2019 of 434,972 [11], a university, an airport, and a harbor. The city is hardly visited by tourists, little contact exists between the local population and Western people. 97% of the population of West-Timor is Christian. Islam is followed by 2.9%, the remaining 0.088% include Hindus and Buddhists [12]. In terms of gross regional product per capita it ranks least among the Indonesian provinces [13]. Stunting rates are high with more than 50% in the rural areas [1].

In this study, we focused on anthropometry, on clinical signs of undernutrition, on biological age, and also on the social, economic, political and emotional (SEPE) [14] situation of the children. We selected the entire body of students of two representative primary state schools; one private school to encompass children of the affluent social strata; and one remotely situated school for children of impoverished suburban dwellers, and political and economic migrants from East Timor with no relevant association to the inner urban social strata of Kupang (the “underprivileged” school).

In addition to our previous study, we included short fitness tests, an assessment of self-confidence, and the child’s estimate of their fathers’ social role. We noted information on water supply, and on television and refrigerator to roughly estimate household prosperity. As we decided to refrain from taking X-rays of the left hand and wrist, we documented the dental state. We did so for estimating biological age, and monitoring developmental tempo, not for decayed, missing and filled teeth (DMF index). Dental age has long been used as a proxy for physical maturation [15,16,17,18]. School grades for language and mathematics were obtained as a rough measure of cognitive development.

Socially, economically, politically and emotionally (SEPE) disadvantaged children are shorter than children from affluent background. In view of our previous work on the lack of association between nutrition and child growth [1], we hypothesized that

The shortness in height of SEPE disadvantaged children is not associated with anthropometric indicators of poor nutrition.SEPE disadvantaged children are delayed in dental development, andSEPE disadvantaged children show lower self-confidence, and consider their fathers’ social role as inferior.A lack of parental school attainment is associated with short stature.

## 2. Sample and Methods

### 2.1. Participants and Design

We measured 723 school children, 354 girls, 369 boys aged 5.83 to 13.83 (mean 9.54) years, from Kupang, West-Timor, Indonesia, in March 2020. These were the entire bodies of students of two representative elementary state schools, situated close to the old harbor (222 boys, 230 girls), one Catholic private school (104 boys, 92 girls) for children of affluent parents who could afford school fees, and one remote run-down school outside the urban area of Kupang (43 boys, 32 girls), some 50 min drive from the center of Kupang. These children were impoverished and appeared different. They were less noisy than the children from the central urban schools, they wore similar, but ragged and dirty school uniforms. Some had no socks or shoes. Several had scarves or healing wounds on feet and lower leg. An additional problem in this school seemed to be the inappropriate communication between teachers and pupils. Many children spoke their local languages with incomplete knowledge of the national Bahasa Indonesia.

Parental informed consent was given. Ethical approval was provided by the Medical and Health Research Ethics Committee. Faculty of Medicine, Public Health and Nursing, Universitas Gadjah Mada-Dr, Sardjito General Hospital; Ref.NO: KE/FK 1440/EC/2019, from 11 December 2019. All individual data were anonymized. We excluded two children because of poorly healed fractures of arm and leg that would have impaired the physical fitness test, one child refused cooperation.

### 2.2. Procedure

All measurements were performed in the presence of the children’s teachers, and supervised and accompanied by altogether 21 local physicians, pediatricians, and medical residents. The children were lightly dressed and measured without shoes. Weight of the school uniforms was found to be close to 300 g in children below age 10 years, and about 400 g in children above age 10 years, and was subtracted from the weight measurements.

Dental state for estimating biological age was defined as the number of erupted permanent teeth in the left lower mandible and examined by the local dentist. School grades of the children in mathematics and language, birth date and information on parent education (the total number of school years of both parents including university education.) were obtained from school records. Information to a condensed “household score” (sum of TV (no = 1, yes = 2), refrigerator (no = 1, yes = 2) and category of drinking water (river water = 1, water from public well = 2, public sanitary facility = 3, tube water = 4, bottle water = 5)) were obtained from the children. 

To estimate nutritional associated status we used body mass index standard deviation scores (BMI_SDS), and mid-upper arm circumference standard deviation score (MUAC_SDS), to estimate fatness and energy balance we used the average of three measurements of triceps and subscapular skinfold thickness. By averaging two skinfolds, we tried to avoid possible confounding influences of SES, ethnic background, age and sex [19,20,21]. In addition, we used two historic indicators of the nutritional state. The “pelidisi” (Pondus dEcies LInear DIviso SedentIs altitude; cubic root of 10 times weight, divided by sitting height) ignored leg length. Livi’s (ponderal) index [22] relates the cubic root of weight to height, and was considered to better mirror the nutritional state than body weight alone with arguments similar to those used today to recommend the BMI. These indices were very popular among German pediatricians, and commonly used, thoroughly discussed and particularly appreciated by Wagner in his early 20th century work on the numerical assessment of the nutritional status [23]. To estimate external skeletal robusticity as a predictor of physical fitness, we measured elbow breadth and calculated the frame index (FI, elbow breadth/height) [24,25,26]. The FI has been used as a proxy for everyday physical activity levels [27]. Physical fitness was also tested by standing long jump and handgrip strength.

Subjective self-confidence and assess of father position were assessed by the MacArthur community ladder, adapted from the MacArthur Scale of Subjective Social Status [28].

The ladder was introduced to the children with the idea that they should consider this ladder as representing their social position among their peers. At the top of the ladder (step 8) are the people who are best regarded, and have the most influence, like friends, and other students of their class. At the bottom of the ladder (step 1) are the people who have the least influence and are regarded the most stupid and the least prestigious within your class. We asked:” where would you place yourself on this ladder” [29,30]? In the same way, we asked the children to assess their father’s position. The top of the ladder was considered to represent very important personalities like the country’s president, two steps below were associated with the social rank of a medical doctor. The lowest level was considered to represent the weakest and most underprivileged members of the society. The intermediate steps were left unassigned for free decisions [31].

Clinical signs of undernutrition and malnutrition (edema, Bitot’s spots, goiter, hair, skin, and general appearance) [32] were checked in the lightly dressed children by local pediatricians. The study included extensive walking through residential areas surveying housing conditions, food markets, and sanitary facilities.

### 2.3. Instruments

Body height and sitting height were determined by digital laser rangefinder GLM Professional^®^ Bosch 250 VF [33] to the nearest millimeter (technical error 2.5 mm), weight by digital scales (Soehnle, Nassau, Germany, Style Sense Compact 100) to the nearest 100 g (technical error 0.15 kg), and skinfold thickness by caliper (Holtain, Ltd. Crosswell, Crymych, UK) to the nearest 0.2 mm (technical error of triceps skinfolds 1.5 mm, technical error of subscapular skinfolds 2.0 mm). All measurements were taken under standardized conditions [34]. The standing long jump was measured with a conventional tape measure. The handgrip strength was measured with Martin-“Virgorimeter” (measurement range 0–160 kP, measurement quality: validity (r = 0.89) [35] and reliability (Intraclass correlation coefficient (ICC) =0.94) [36]). We used the ball with a diameter of 4 cm as suggested for children aged 4 to 10.

### 2.4. Data Analysis

Because several variables depend on sex and age, we transformed height, weight, BMI, MUAC, average triceps and subscapular skinfold thickness, hand grip strength, standing long jump, Ladder-test, sum of erupted teeth and household score into standard deviation scores (SDS). SDS for height (hSDS) and body mass index (BMI_SDS) were calculated according to WHO references [37]. SDS of all other variables were calculated from the sample of the schoolchildren. The schools were categorized as private school, state schools (included pupils of two schools) and the “underprivileged” school.

We used *t*-test for comparing the data of the private and the underprivileged schoolchildren. Nonparametric correlations were checked. In order to better understand the influence of the various variables on child growth, we excluded age, height, and also weight and elbow breadth (information is included in BMI respectively FI), and applied principal component analysis (PCA) with Varimax-rotation on the remaining variables. The Kaiser-Meyer-Olkin-criteria were checked (KMO > 0.75). Using screen plots and eigenvalue criteria five principal components (PC) were chosen respectively which explained 62% of variance. We determined the variables with the highest loading value in each PC and referring to these variables, named the components anthropometry, household, education, sports, and self-perception. We then used the individual PC in multiple regression (LM with backward selection) against hSDS. 

The descriptive statistic and the Linear Models were performed with the programming language “R” (R-version 3.5.1 2018), the PCA were done with SPSS version 26 (IBM SPSS Statistics, Armonk, NY, USA).

## 3. Results

The children of Kupang, West-Timor, Indonesia, are short and thin (Figure 1).

The prevalence of stunting in boys was 8.5%, 26.4%, and 46.8% (private school, state schools, underprivileged school). The prevalence of stunting in girls was 10.4%, 20.3%, and 25.5% (private school, state schools, underprivileged school). Descriptive statistics of all schoolchildren are given in Table 1. Private school children are tallest, heaviest, and have highest BMI; they have largest MUAC; their parents are best educated, their living conditions judged by presence of television, refrigerator, and water quality are best. They have best school grades in language, but the differences between affluent and underprivileged children were small and insignificant for mathematics in the girls. Girls of the underprivileged school were less stunted than boys from that school. The differences private vs. state, and state vs. underpriviliged school children, remain were also significant, except for frame index in both sexes, school achievements of private vs. state school boys, and height SDS, BMI_SDS, and MUAC_SDS of state vs. indigent school girls.Typical clinical signs of under- or malnutrition were absent. The frequent traumatic skin infections that we often detected on feet and ankles of the underprivileged children, were obviously due to lack of shoes and socks.

Table 2 shows numbers and standard deviations of erupted left lower mandible teeth. We found a mild retardation of the dental development in the underprivileged children that however, did not reach the level of significance. Thus, even in the underprivileged children we were unable to attribute the shortness in stature to any significant delay in developmental tempo.

Underprivileged children are physically fitter. They showed best results in standing jump, and their high skeletal robusticity (highest Frame index) suggested persistent high levels of daily physical activity. Underprivileged children are short, but not thin. The correlations between height and the measures of the nutritional state were generally low in most children of Kupang, but almost completely disappeared in the children of the underprivileged school (Table 3, Figure 2).

Testing for self-confidence using the ladder-test yielded unexpected results. In spite of their impoverished living conditions, the ladder test did not indicate reduced self-confidence in the underprivileged children. We also failed to detect any measurable difference in the children’s perception of father’s social status indicating that even the underprivileged children were mainly unaware of their economically unfavorable living conditions and social position.

Handgrip strength coincided mildly with height and the nutritional state, standing long jump showed no relation with height, and a revers relation in the heavier wealthy children.

Table 4 depicts the correlation between parental education, and the mathematics and language grades of the children on height. The correlations are generally weak. Maternal education appears of importance for body height both in the affluent and in the underprivileged children. Language faculty appears particularly important in the underprivileged children possibly reflecting incomplete knowledge of the national Indonesian language.

We found weak negative associations between educational variables and self-perception in the underprivileged children. Also the associations between self-confidence and the perceived social status of the father were negative (boys r = −0.24; girls r = −0.37; *p* < 0.001) in all schools (Figure 1).

In order to better focus on the interaction of the 18 variables obtained in this study, and their influence on height, we excluded age and height, and also weight and elbow breadth (information is included in BMI respectively FI) and applied Principal Component Analysis (PCA) to the remaining variables. Similar to what was already visible in Figure 2 the rotated component matrix showed five major Principal Components (Table 5) explaining 62% of the variance: a combination of anthropometric variables (component: anthrop) consisting of the indicators of the nutritional status BMI, MUAC_SDS and skinfold_SDS, and the frame index (FI), that explained 23.34% of the variance; water quality, and presence of TV and refrigerator (component: household) that explained 13.68% of the variance highlighting the impact of household conditions. Parental education and school grades (component: education) contributed to 8.89 percent, handgrip strength and standing long jump to 8.8% (component: sport), and the ladder test to less than 8% (component: perception). We excluded the component “perception” because of the low variance in the PCA. We then used the remaining individual PC in multiple regression (LM with backward selection) against hSDS and further differentiated the results at the level of the school, by multiple regression with backward selection (data not shown in detail). The influence of the component “anthrop” on body height declined from private to underprivileged school from β = 0.58 (*p* < 0.000) to β = −0.3 (*p* = 0.782) in boys and from β = 0.47 (*p* < 0.000) to β = 0.52 (*p* = 0.021) in girls. The influence of component “education” was strongest in girls of the underprivileged school (β = 0.74; *p* = 0.001) and weak in the state schoolchildren (β = 0.33; *p* = 0.000), and in the boys of the private school (β = 0.2; *p* = 0.034). The analysis confirmed the findings, and highlighted the significance of parental education on growth in the girls of the underprivileged school. The component household had no particular impact on body height in either school.

## 4. Discussion

### 4.1. Summary of Main Results

Indonesian schoolchildren are short compared to WHO reference [37]. This also applies for the children of Kupang, West-Timor, Indonesia. In this study mean hSDS was −0.63 and stunting prevalence was 8.5% in the affluent boys, and mean hSDS −0.74 and stunting prevalence was 10.4% in the affluent girls. Stunting prevalence in the children of impoverished suburban dwellers, and political and economic migrants with no relevant association to the inner urban social strata of Kupang was 46.8% in the boys, and 25.5% in the girls. The sex difference reflects the higher susceptibility of boys to disadvantageous living conditions [38]. Children from affluent background were not only tallest, but had also highest BMI, and strongest MUAC, confirming general knowledge since more than a century [4]. Yet as shown in Figure 2, the obvious association between the indicators of nutritional status and body height depended on the social background, it was strongest in the private school children, it was lower in the state schoolchildren and lowest in the children of the underprivileged school. The observation strongly suggests that the shortness in height of SEPE disadvantaged children is not associated with indicators of poor nutrition, and confirming our first hypothesis.

### 4.2. Interpretation of the Main Results

Why are the Indonesian school children so short?

In the pediatric practice, being short and being delayed in physical development and maturation are often associated. Children with constitutional delay of growth and puberty are short [39]. We had decided to refrain from taking X-rays of the left hand and wrist, and instead, documented the dental state for estimating biological age [17]. We hypothesized that the shortness particularly of the SEPE disadvantaged children is associated with delay in biological age and physical development and maturation as indicated by delayed dental development. However, this was not the case. Children of the underprivileged school are not short because of delay in their dental state and we rejected the second hypothesis (Table 1). 

Accordingly, the association between being tall and being accelerated in physical development and maturation has also frequently been observed. In addition, the association between being tall and being obese is common and considered to be due to early estrogenization and the action of insulin on the IGF-1 receptor [40]. Figure 1 indicates that a non-negligible percentage of children in private and also in the state schools were obese, with skinfolds above 2 SDS for skinfold thickness. The lack of association between height and indicators of the nutritional status in the underprivileged children suggests that affluent children appear “too tall” due to their mild acceleration of the developmental tempo, rather than that the underprivileged children appear too short. The findings strongly negate nutritional reasons for being responsible for the shortness of the disadvantaged schoolchildren, and contrast the common notion that “the use of length-for-age as the indicator of choice in monitoring the long-term impact of chronic nutritional deficiency” [41]. Except for the general impression that the underprivileged children were less noisy and appeared more obedient to instructions of their teachers, we found no evidence that these children perceived themselves as subordinate. 

Quite in contrast to our initial study concept assessing subjective self-confidence by the MacArthur community ladder [28], we observed that most of the West-Timor children were unable to appropriately place themselves on this symbol for Western-style vertically structured social systems [29,30]. Most of these children tended to considered themselves at the top of the ladder, and except for some of the older private school children, maintained this–in our eyes unrealistic–self-perception up to age 12 to 13 years. Böker et al. [31] discussed these findings and emphasized that the concept of ranking oneself “high” and prestigious, or “low”, or “climing up and down”on a social ladder is strongly influenced by culture. Even within American studies the MacArthur test appears to work best for White and Chinese Americans [42] and less well for Latino and African Americans [42,43]. This tendency was generally confirmed in a study, in which Western and non-Western societies were compared [30]. Thus, our hypothesis that SEPE disadvantaged children tested by vertical ranking, show lower self-confidence, and consider their fathers’ social role as inferior, had to be rejected as our Western cultural perspectives appear inappropriate for school children in West-Timor, Indonesia.

### 4.3. Implications to Parents, Policymakers, Administrators or Clinicians

Indonesian schoolchildren are short but their shortness is not a matter of nutrition. Education is important. The influence on growth, of parental education and individual school grades was strongest in girls of the underprivileged school and weaker in the state school children, and in the private school boys. This underlines the significance of parental education particularly in the girls of the poorest socio-economic background. The findings are in line with Jeong et al. [44] who had summarized evidence that maternal and paternal education were independently associated with 0.37 (95% CI 0.33 to 0.41) and 0.20 (95% CI 0.16 to 0.24) higher height-for-age z-scores, and 0.31 (95% CI 0.29 to 0.34) and 0.16 (95% CI 0.14 to 0.18) higher Early Childhood Development Index z-scores, respectively (comparing secondary or higher to no education). The associations were stronger for maternal education than paternal education but comparable between child outcomes, whereas the factor household prosperity even though associated with parental education as shown by Vaivada et al. [45], appeared to have no direct impact on height in the children of either school type.

### 4.4. Strengths and Limitations

We measured and analyzed children from all social and economic strata. The data were obtained with the help and in the presence of local physicians, medical students, and teachers of the schools ensuring best possible conditions for data acquisition and cooperation of the children. Except for three children who were excluded because of poorly healed fractures that would have impaired the physical fitness test, and one child who refused cooperation, we were able to measure complete bodies of school children. With 104 boys and 92 girls from a Catholic private school, the number of affluent children may be slightly overrepresented when compared to the Indonesian population [46]. A limitation of the present study may be the fact that due to very limited migration in recent history, the population of Kupang is rather homogeneous, and different from the majority of the Indonesian population, mostly Christian. Thus the present findings may not be fully representative for the large and genetically and religiously heterogeneous Indonesian population.

## 5. Conclusions

The present study confirms historic findings published already at the beginning of the 20th century that malnutrition cannot be considered “as the decisive factor in the undersize of the poor.” Socially, economically, politically and emotionally (SEPE) disadvantaged children are shorter than children from affluent background. Their shortness was not associated with clinical signs of malnutrition, nor with thinness, nor with delay in physical development. Quite in contrast, the association between body height and anthropometric indicators of the state of nutrition such as BMI, MUAC, skinfold thickness and two historic indicators of undernutrition, was weakest in the disadvantaged children, and strongest in the wealthy suggesting that the shortness in height of SEPE disadvantaged children is not associated with poor nutrition. Body height was associated with parental education. Our hypothesis that SEPE disadvantaged children show lower self-confidence, and consider their fathers’ social role as inferior, had to be rejected as our Western cultural concepts appear inappropriate when testing school children in West-Timor, Indonesia.

Stunting is not a synonym of malnutrition, stunting is a synonym of social disadvantage and poor parental education.

## Figures and Tables

**Figure 1 ijerph-18-01350-f001:**
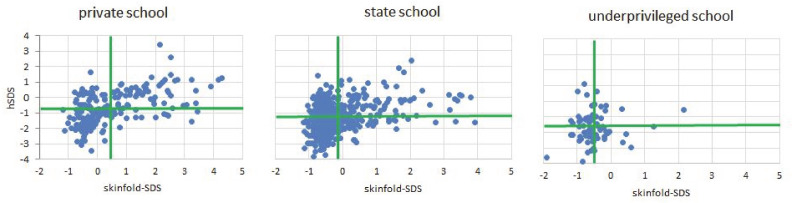
Height (hSDS) and skinfold thickness (skinfold-SDS) of the children from the private school, state schools, and the underprivileged school in Kupang (West-Timor, Indonesia). Green bars indicate mean values of hSDS and skinfold SDS.

**Figure 2 ijerph-18-01350-f002:**
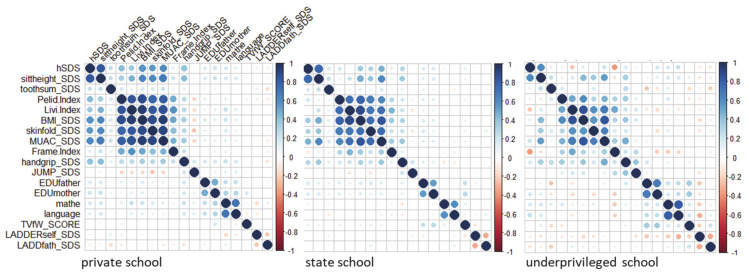
Correlation matrices of 18 variables of three different school types in Kupang (West-Timor) depicts the correlation matrices of the private, the state, and the underprivileged schoolchildren and highlights the differences between the school types. High correlations existed between the two measures of length (total body height and sitting height), and between the five measures related to the nutritional state. The correlations between height and the measures of the nutritional state however, were low and strongly depended on the school type (Table 3). The correlations between height SDS and variables related to the nutritional state seemed to almost completely disappear in the children of the underprivileged school. Also the interaction of the five measures of the nutritional state significantly declined in the underprivileged school children.

**Table 1 ijerph-18-01350-t001:** (**a**) Descriptive statistics (mean, SD) boys of private school (*n* = 104, state schools (*n* = 222), underprivileged school (*n* = 43) and (**b**) girls of private school (*n* = 92), state schools (*n* = 230), underprivileged school (*n* = 32) in Kupang (West-Timor). Significant *p*-values (*p* ≤ 0.05) of the differences between private and underprivileged school children are indicated by bold letters, 95% confidence interval are added (CI 95%)).

**(a)**									
**Boys**	**Private**		**State**		**Underprivileged**		**Difference Private and Underprivileged**		
	**Mean**	**SD**	**Mean**	**SD**	**Mean**	**SD**	***p***	**CI 95%**	
height SDS	−0.63	1.2	−1.37	0.89	−1.92	0.79	**<0.000**	−1.685	−0.899
sittheight_SDS	0.54	1.14	−0.11	0.84	−0.73	0.62	**<0.000**	−1.632	0.541
toothsum_SDS	0.04	0.95	0.01	0.96	−0.16	0.91	0.225	−0.544	0.129
BMI_SDS	0.09	1.88	−1.11	1.53	−1.66	1	**<0.000**	−2.347	−1.150
Livi.Index	2.35	0.15	2.27	0.12	2.25	0.07	**<0.000**	−0.150	−0.055
Pelid.Index	96.2	6.62	93.28	5.34	92.48	2.64	**<0.000**	−5.788	−1.661
MUAC_SDS	0.56	1.28	−0.17	0.78	−0.48	0.44	**<0.000**	−1.433	−0.643
skinfold_SDS	0.59	1.31	−0.18	0.76	−0.5	0.29	**<0.000**	−1.481	−0.682
Frame.Index	38.32	2.01	38.73	1.85	39.01	1.68	**0.050**	−0.001	1.377
handgrip_SDS	0.05	1.11	0.02	0.92	−0.21	0.95	0.175	−0.647	0.119
JUMP_SDS	−0.44	0.97	0.19	0.96	0.08	0.9	**0.003**	0.181	0.860
EDUfather	12.52	2.84	11.22	2.81	8.44	2.88	**<0.000**	−5.152	−3.026
EDUmother	12.38	2.7	11.14	2.57	7.9	2.99	**<0.000**	−5.482	−3.461
math (grade)	76.02	13.81	73.22	11.1	68.63	7.51	**0.002**	−11.954	−2.834
language (grade)	77.82	10.35	76.57	10.38	70.48	7.12	**<0.000**	−10.864	−3.820
TVfW_SCORE	6.26	0.64	6.01	0.99	4.72	1.26	**<0.000**	−1.849	−1.229
LADDERself_SDS	0.02	0.94	−0.02	1.04	0.28	1.33	0.184	−0.124	0.640
LADDERfath_SDS	0.23	0.81	−0.1	1.04	−0.04	1.06	0.098	−0.589	0.050
**(b)**									
**Girls**	**Private**		**State**		**Underprivileged**		**Difference Private and Underprivileged**		
	**Mean**	**SD**	**Mean**	**SD**	**Mean**	**SD**	***p***	**CI 95%**	
height SDS	−0.74	0.99	−1.17	1.01	−1.34	1.1	**0.005**	−1.023	−0.193
sittheight_SDS	0.26	0.99	−0.03	0.97	−0.55	0.94	**<0.000**	−1.213	−0.416
toothsum_SDS	−0.15	1.1	−0.01	0.97	−0.43	0.8	0.184	−0.700	0.136
BMI_SDS	−0.2	1.31	−0.84	1.35	−1.25	1.27	**<0.000**	−1.576	−0.520
Livi.Index	2.32	0.12	2.27	0.11	2.25	0.11	**0.006**	−0.116	−0.019
Pelid.Index	95.06	4.92	93.18	4.48	93.3	3.72	**0.068**	−3.646	0.128
MUAC_SDS	0.4	1.1	−0.12	0.92	−0.32	0.84	**0.001**	−1.148	−0.301
skinfold_SDS	0.35	1.06	−0.07	0.95	−0.51	0.75	**<0.000**	−1.256	−0.451
Frame.Index	36.89	1.98	37.48	1.74	37.75	1.88	**0.035**	0.062	1.652
handgrip_SDS	−0.1	1.09	0.1	0.96	−0.22	0.74	0.538	−0.538	0.282
JUMP_SDS	−0.51	0.75	0	0.97	0.55	1	**<0.000**	0.726	1.391
EDUfather	12.49	3.14	11.19	2.83	9.16	3.73	**<0.000**	−4.693	−1.983
EDUmother	12.21	3.43	11.11	2.65	9.44	3.15	**<0.000**	−4.154	−1.394
math (grade)	75.34	12.36	75.07	10.36	71.96	6.24	0.167	−8.189	1.436
language (grade)	79.89	8.56	79.55	9.87	73.64	6.98	**0.001**	−9.765	−2.729
TVfW_SCORE	6.29	0.58	5.86	0.99	4.81	1.31	**<0.000**	−1.811	−1.135
LADDERself_SDS	0.03	0.9	0.03	1.07	−0.05	1.21	0.717	−0.475	0.328
LADDERfath_SDS	0.22	0.75	−0.11	1.08	0.16	0.86	0.742	−0.368	0.263

**Table 2 ijerph-18-01350-t002:** Mean number (N) and standard deviation (SD) of erupted left lower mandible teeth (dental state) in the school children of Kupang (West-Timor).

Age (Years)	Girls		Boys	
	N	SD	N	SD
6	2.1	1.1	1.5	0.9
7	3	0.9	2.4	0.9
8	3.7	0.9	3	0.9
9	4.7	1.4	4.1	1.1
10	5.8	1.2	5.1	1.2
11	6.5	0.9	6.1	1.1
12	6.8	0.8	6.4	1

**Table 3 ijerph-18-01350-t003:** Correlation coefficients between height SDS and measures of the nutritional state according to school type in Kupang (West-Timor). r < 0.01 are indicated by bold numbers.

School	Girls			Boys		
	Private	State	Underprivileged	Private	State	Underprivileged
Pelidisi	**0.28**	**0.33**	0.02	**0.55**	0.16	0.08
Livi	0.19	0.18	−0.10	**0.52**	0.06	−0.29
BMI_SDS	**0.44**	**0.47**	0.16	**0.65**	**0.29**	0.07
skinfold_SDS	**0.43**	**0.40**	0.13	**0.61**	**0.28**	0.09
MUAC_SDS	**0.49**	**0.54**	**0.32**	**0.69**	**0.42**	0.22

**Table 4 ijerph-18-01350-t004:** Correlation coefficients between height SDS and the educational variables according to private and underprivileged school in Kupang (West-Timor). r < 0.05 are indicated by bold numbers.

School	EDUfather	EDUmother	Math Grades	Language Grades
Private	0.162	**0.205**	**0.181**	0.141
underprivileged	0.060	0.199	0.186	**0.241**

**Table 5 ijerph-18-01350-t005:** Rotated component matrix of Principal Component Analysis (PCA) after excluding age, height, weight, and elbow breadth. The remaining variables can be separated into five major Components, Varimax-Rotation, Eigenvalues > 0.5 are bold.

Component	Anthrop	Household	Education	Sport	Perception
% of variance	23.34	13.68	8.89	8.80	7.29
BMI_SDS	**0.917**	−0.064	0.151	0.057	0.003
Frame.Index	**0.500**	−0.014	−0.236	0.325	0.039
EDUfather	0.077	−0.295	**0.760**	0.093	−0.037
EDUmother	0.100	−0.306	**0.773**	0.047	−0.090
average school grades	0.075	0.093	**0.584**	−0.086	0.194
Water	−0.037	**0.723**	−0.020	−0.046	−0.150
TV	−0.004	**0.697**	−0.130	0	−0.003
Fridge	−0.119	**0.717**	−0.168	0.079	0.012
toothsum_SDS	0.205	−0.136	−0.093	0.428	−0.146
MUAC_SDS	**0.940**	−0.059	0.147	0.067	−0.018
skinfold_SDS	**0.911**	−0.080	0.136	−0.036	0.013
handgrip_SDS	0.289	0.074	0.199	**0.683**	−0.055
JUMP_SDS	−0.316	0.111	−0.023	**0.758**	0.151
LADDERself_SDS	−0.050	−0.092	0.024	0.059	**0.785**
LADDERfath_SDS	0.064	−0.038	0.043	−0.095	**0.780**

## Data Availability

Data supporting reported results can be viewed on request, from the first author Dr. C. Scheffler.

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
