# Peer review of "Stunting as a Synonym of Social Disadvantage and Poor Parental Education"

_ijerph, 2021, doi:10.3390/ijerph18031350_

Round 1

Reviewer 1 Report

This is a well thought out and well designed study. In the Introduction section the authors state, 'Catch-up growth is cause-specific and characterized by weight and height velocity above the limits of normal for age"... can authors explain more what they mean? The hypothesis is clearly stated. The methodology is sound. The authors have used dental age as a proxy for biologic age - is this the method used widely? Are there additional supporting data (in addition to ref. 35)? The study is conducted in a defined school age population 2 schools. Can the authors elaborate on how the findings can be generalized and applied population wide? It would be desirable for the investigators to comment on limitations of the study.

Reviewer 2 Report

Paper by Doctor Sheffler C et al. treated the association between stunting and socioeconomic status in Indonesia. I would like to provide comments to improve the manuscript.

[Major]

  1. Abstract lacks arithmetic figures of the main results. It may cause less impact of this study.
  2. Introduction, especially the first paragraph is lengthy. As a reader, I would like to be explained background and the rationale of the study briefly. This is associated with readability and therefore utility of this study.
  3. At the end of the Introduction section, there may be methodology (i.e. “we included short fitness tests, …”). If it is true, the paper need to divide the sections, move the methodology to the Methods section, and describe compactly.
  4. Limitation section need to be added to the Discussion section. For example, is there bias associated with investigation only in West Timor? Did the researchers consider genetic dispositions, when they distribute the results to the people in the world?
  5. Sample and Methods section may be relatively lengthy. More brief description is needed.
  6. Table 1 may be lengthy.
  7. I consider that p value need to be calculated both for “difference between state vs. private” and “poor vs. private”.
  8. Table 2: In odontologic epidemiology, decayed, missing and filled teeth (DMF index) may be more commonly used.
  9. Figure 2, table 3 and table 5 may not be necessary for academic paper. They are just correlations, and could not results in the main text. Rather than them, association of anthropometric measures and their SDS scores with socioeconomic status need to be more highlighted.
  10. The Discussion section need to be divided to 1. Summary of the main results for the purpose: 2. Interpretation of the main results in context with the previous study results; 3. Implications to parents, policymakers, administrators or clinicians; 4. Strengths and limitations of the methodology; 5. Simple conclusion(s). The discussion in the present style may reduce readability.
  11. I would like to request more analyses of association of anthropometric measures and their SDS scores with socioeconomic statuses. I consider the analyses would correspond to the purpose of this study.

[Minor]

  1. Discussion: “Indonesian school children are short” need to be more explained. Do the authors compare them with whom?

  1. The name of “poor school” may assemble discrimination. It may need to be renamed, because this is academic paper.

  1. Table 1: p value need to be described as “<0.0001”, for example, in place of “0.000”.

I believe that with modifications, this study would add great evidence to the fields of social epidemiology and child health.

Reviewer 3 Report

Review of the manuscript "Stunting as a synonym of social disadvantage and poor parental education".

After reviewing your manuscript, I consider your efforts to have been very successful and consider this article to be scientifically sound. However, I would like to suggest a few recommendations prior to publication. In order:

Key words: I suggest to eliminate the following "political and emotional factors on growth" because it is not exactly a key word. Authors should find a shorter word in repositories suitable for the topic.

I suggest authors to change the title of point 2 "Sample and Methods" to Material and Methods. I also suggest that they add the corresponding subsections: Participants and design; instruments; procedure; data analysis.

In the results section, authors should be careful when referencing figures and tables. I remind you that the tables have the title at the top and the notes at the bottom of the table. In turn, the figures are referenced in the footer of the figure. On both occasions the font size is smaller than the rest of the text.

Once these modifications have been made, I suggest the publication of the manuscript presented.

Round 2

Reviewer 2 Report

Thank you for revising the manuscript. I have no more concern. I appreciate your efforts to report the important results.